# DYNAMIC RELATIONAL INFERENCE IN MULTI-AGENT TRAJECTORIES

## ABSTRACT

Unsupervised learning of interactions from multi-agent trajectories has broad applications in physics, vision and robotics. However, existing neural relational inference works are limited to *static* relations. In this paper, we consider a more general setting of dynamic relational inference where interactions change over time. We propose DYnamic multi-Agent Relational Inference (DYARI) model, a deep generative model that can reason about *dynamic* relations. Using a simulated physics system, we study various dynamic relation scenarios, including periodic and additive dynamics. We perform comprehensive study on the trade-off between dynamic and inference period, the impact of training scheme, and model architecture on dynamic relational inference accuracy. We also showcase an application of our model to infer coordination and competition patterns from real-world multi-agent basketball trajectories.

## 1 INTRODUCTION

Particles, friends, and teams are multi-agent relations at different scales. Learning multi-agent interactions is essential to our understanding of the structures and dynamics underlying many systems. Practical examples include understanding social dynamics among pedestrians (Alahi et al., 2016), learning communication protocols in traffic (Sukhbaatar et al., 2016; Lowe et al., 2017) and predicting physical interactions of particles (Mrowca et al., 2018; Li et al., 2018; Sanchez-Gonzalez et al., 2020). Most existing work on modeling relations assume the interactions are *observed* and train the models with *supervised* learning. For multi-agent trajectories, the interactions are *hidden* and thus need to be inferred from data in an *unsupervised* fashion. While one could impose an interaction graph structure (Battaglia et al., 2016), it is difficult to find the correct structure as the search space is very large (Grosse et al., 2012). The search task is computationally expensive and the resulting model can potentially suffer from the model misspecification issue (Koopmans & Reiersol, 1950).

Relational inference aims to discover hidden interactions from data and has been studied for decades. Statistical relational learning are based on probabilistic graphical models such as probabilistic relational model (Kemp & Tenenbaum, 2008; Getoor et al., 2001; Koller et al., 2007; Shum et al., 2019). However, these methods may require significant feature engineering and high computational costs. Recently, Battaglia et al. (2016); Santoro et al. (2017) propose to reason about relations using graph neural networks but still requires supervision. One exception is Neural Relational Inference (NRI) (Kipf et al., 2018), a flexible deep generative model that can infer potential relations in an unsupervised fashion. As shown in Figure 1, NRI simultaneously learns the dynamics from multi-agent trajectories and infers their relations. In particular, NRI builds upon variational auto-encoder (VAE) (Kingma & Welling, 2013) and introduces latent variables to represent the hidden relations. Despite its flexibility, a major limiting factor of NRI is that it assumes the relations among the agents are *static*. That is, two agents are either interacting or not interacting regardless of their states at different time steps, which is rather restrictive.

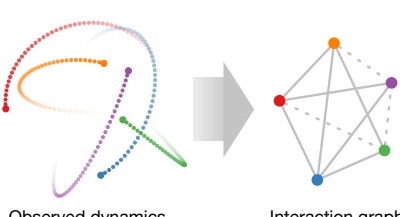

Observed dynamics          Interaction graph

Figure 1: Neural Relational Inference for learning the interaction graph. Picture taken from (Kipf et al., 2018)

In this paper, we study a more realistic setting: *dynamic* relational inference. For example, in game plays, players can coordinate and compete dynamically depending on the strategy. We propose a novel deep generative model, which we call DYnamic multi-Agent Relational Inference (DYARI). DYARI encodes trajectory interactions at different time steps. It utilizes deep temporal CNN models with pyramid pooling to extract rich representations from the interactions. DYARI infers the relations for each sub-sequence dynamically and jointly decode a sequence of relations.

As relational inference is unsupervised, we use simulated dynamics physics systems as ground truth for validation. We find that the performance of the static NRI model deteriorates significantly with shorter output trajectories, making it unsuitable for dynamic relational inference. In contrast, DYARI is able to accurate infer the hidden relations with various dynamics scenarios. We also perform extensive ablative study to understand the effect of inference period, training schemes and model architecture. Finally, We showcase our DYARI model on real-world basketball trajectories.

In summary, our contributions include:

- We tackle the challenging problem of unsupervised learning of hidden dynamic relations given multi-agent trajectories.
- We develop a novel deep generative model called DYARI to handle time-varying interactions and predict a sequence of hidden relations in an end-to-end fashion.
- We demonstrate the effectiveness our method on both the simulated physics dynamics and real-world basketball game play datasets.

## 2 RELATED WORK

**Deep sequence models**    Deep sequence models include both deterministic models (Alahi et al., 2016; Li et al., 2019; Mittal et al., 2020) and stochastic models (Chung et al., 2015; Fraccaro et al., 2016; Krishnan et al., 2017; Rangapuram et al., 2018; Chen et al., 2018; Huang et al., 2018; Yoon et al., 2019). For GAN-like models, (Yoon et al., 2019) combine adversarial training and a supervised learning objective for time series forecasting. Liu et al. (2019) propose a non-autoregressive model for sequence generation. Compared with GANs, VAE-type models can provide explicit inference and are preferable for our purpose. For instance, Chung et al. (2015) introduces stochastic layers in recurrent neural networks to model speech and hand-writing. Rangapuram et al. (2018) parameterizes a linear state-space model for probabilistic time series forecasting. Chen et al. (2018); Huang et al. (2018) combine normalizing flows with autoregressive models. However, all existing models only model the *temporal* latent states for individual sequences rather than their *interactions*.

**Relational inference**    Graph neural networks (GNNs) seek to learn representations over relational data, see several recent surveys on GNNs and the references therein, e.g. (Wu et al., 2019; Goyal & Ferrara, 2018). Unfortunately, most existing work assume the graph structure is *observed* and train with supervised learning. In contrast, relational inference aims to discover the *hidden* interactions and is unsupervised. Earlier work in relational reasoning (Koller et al., 2007) use probabilistic graphical models, but requires significant feature engineering. The seminal work of NRI (Kipf et al., 2018) use neural networks to reason in dynamic physical systems. Alet et al. (2019) reformulates NRI as meta-learning and proposes simulated annealing to search for graph structures. Relational inference is also posed as Granger causal inference for sequences (Louizos et al., 2017; Löwe et al., 2020). Nevertheless, all existing work are limited to *static* relations while we focus on *dynamic* relations.

**Multi-agent learning**    Multi-agent trajectories arises frequently in reinforcement learning (RL) and imitation learning (IL) (Albrecht & Stone, 2018; Jaderberg et al., 2019). Modeling agent interactions given dynamic observations from the environment remains a central topic. In the RL setting, for example, Sukhbaatar et al. (2016) models the control policy in a fully cooperative multi-agent setting and applies a GNN to represent the communications. Le et al. (2017) models the agents coordination as a latent variable for imitation learning. Song et al. (2018) generalizes GAIL (Ho & Ermon, 2016) to multi-agent through a shared generator. However, these coordination models only capture the global interactions implicitly without the explicit graph structure. Tacchetti et al. (2019) combines GNN with a forward dynamics model to model multi-agent coordination but also requires supervision. Grover et al. (2018) directly models the episodes of interaction data with GNs for learning multi-agent policies. Our method instantiates the multi-agent imitation learning framework, but focuses on relational inference. Our approach is also applicable to dynamic modeling in model-based RL.

## 3 DYNAMIC MULTI-AGENT RELATIONAL INFERENCE

Given a collection of multi-agent trajectories, we aim to reason about their hidden relations over time. First we describe the underlying probabilistic inference problem.

### 3.1 PROBABILISTIC INFERENCE FORMULATION

For each agent $i \in \{1, \cdots, N\}$, define its state (coordinates) as $x_t \in \mathbb{R}^D$. A trajectory $\tau^{(i)} = (x_1, x_2, \cdots, x_T)$ is a sequence of states that are sampled from a policy. Given trajectories from $N$ agents $\{\tau^{(i)}\}_{i=1}^N$, dynamic relational inference aims to infer the pairwise interactions of $N$ agents at every time step. Mathematically speaking, the joint distribution of the trajectories can be written as:

$$p(\tau^{(1)}, \cdots, \tau^{(N)}) = \prod_{t=1}^{T} p(\mathbf{x}_{t+1}|\mathbf{x}_t, \cdots, \mathbf{x}_1) \qquad (1)$$

where $p(\mathbf{x}_{t+1}|\mathbf{x}_t, \cdots, \mathbf{x}_1)$ represents the state transition dynamics. We use the bold form $\mathbf{x}_t := (x_t^{(1)}, \cdots, x_t^{(N)})$ to indicate the concatenation of all agents observations and $\mathbf{x}_{<t} := (\mathbf{x}_1, \cdots, \mathbf{x}_t)$.

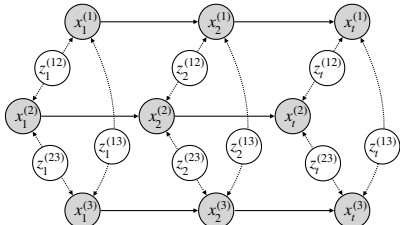

We introduce latent variables $z_t^{(ij)}$ to denote the interactions between agent $i$ and $j$ at time $t$. To make the problem tractable, we restrict $z_t^{(ij)}$ to be categorical, representing discrete interactions such as coordination or competition. We assume that the dynamics model can be decomposed into the individual dynamics, in conjunction with the pairwise interaction. This substantially reduces the dimensionality of the distribution and simplifies learning. Therefore, we can rewrite the transition dynamics as:

Figure 2: Probabilistic graphical model representation of dynamic multi-agent relational inference. Shaded nodes are observed and unshaded nodes are hidden variables.

$$p(\mathbf{x}_{t+1}|\mathbf{x}_{<t}) \approx \int_{\mathbf{z}} \prod_{i=1}^{N} p(x_{t+1}^{(i)}|x_{<t}^{(i)}, z_t^{(ij)}) \prod_{i=1}^{N} \prod_{j=1, j\neq i}^{N} p(z_t^{(ij)}|x_{<t}^{(i)}, x_{<t}^{(j)}) d\mathbf{z} \qquad (2)$$

Here each $p(x_{t+1}^{(i)}|x_{<t}^{(i)})$ captures the state transition dynamics of a single agent. $p(z_t^{(ij)}|x_{<t}^{(i)}, x_{<t}^{(j)})$ represents the latent interactions between two agents. Figure 2 visualizes the graphical model representation for three agents over $t$ number of time steps. The shaded nodes represent observed variables and the unshaded nodes are latent variables. Dynamic relational inference is to estimate distributions of the hidden variables $\{z_t^{(ij)}\}$ at different time steps.

### 3.2 DYNAMIC MULTIAGENT RELATIONAL INFERENCE (DYARI)

We propose a deep generative model: Dynamic multi-Agent Relational Inference (DYARI). Given the trajectories $(\mathbf{x}_1, \cdots, \mathbf{x}_T)$ of all agents, DYARI first concatenates the trajectories based on a fully connected graph. The concatenated trajectories are used as interaction features for the encoder. Then we sample the sequence of relations from the encoded hidden states. Finally, we generate the future trajectory predictions conditioned on the sampled relations. Figure 3 visualizes the overall architecture of our model which encodes and decodes multi-agent trajectories. The bottom cut-out diagram shows the architecture of our encoder.

**Encoder.**   A key ingredient of DYARI is an encoder that is inspired by PSPNet (Zhao et al., 2017) to learn rich representations of trajectories at different scales. In particular, we define a residual block as a two-layer CNN with residual connections (He et al., 2016). Our encoder has four modules: feature extraction, pyramid pooling, an aggregation module, and an interpolation module.

- *Feature extraction*: the feature extraction module consists of multiple residual blocks interleaved with pooling layers to extract rich temporal features.
- *Pyramid pooling*: the pyramid pooling module learns multi-scale temporal representations from the extracted features. First, the output of the feature extraction module is downsampled by 2x

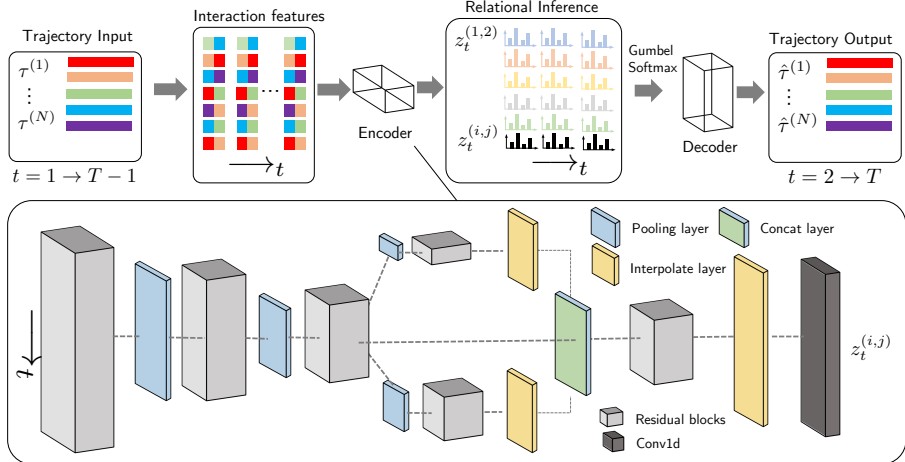

Figure 3: Visualization of the DYARI model. It infers pairwise relations at different time steps given trajectories. The bottom diagram shows the details of the encoder and the decoder is the same as NRI.

and 5x through average pooling. Then, the downsampled features are passed through two residual blocks and finally upsampled by 2x and 5x to generate features which are of the same size as the input. The representations learned at 2x and 5x resolutions are concatenated with the input to generate composite multi-scale features.

- *Aggregation module*: a 1-D convolution that aggregates the multi-scale features.
- *Interpolation module*: it average-pools the aggregated features corresponding to the dynamic period. Then the outputs are upsampled through nearest neighbours interpolation to obtain the hidden presentations for the relations.

**Sampling.** We utilize variational inference (Kingma & Welling, 2013) to sample the latent variables from hidden representations. Specifically, assume the interaction posterior $z_t^{(ij)}$ to be categorical:

$$q_\phi(z_t^{(ij)}|\mathbf{x}_{<t}) \sim \text{Cat}(p_1, \cdots, p_k)$$

Using the Gumbel-Max trick (Jang et al., 2017), we can reparameterize the categorical distribution as: $z_t^{(ij)} = \texttt{Softmax}(h_t^{(ij)} + g_t^{(ij)})$. Here $h_t^{(ij)}$ is the hidden states of the encoder and $g_t^{(ij)}$ is a random Gumbel vector. Note that a defining feature of DYARI is that the latent variable $z_t^{(ij)}$ is time-dependent, requiring fine-grained modeling. Our encoder ensures that the learned representations are expressive enough to capture such complex dynamics.

**Decoder.** Given the sampled latent variables, the decoder generates the prediction auto-regressively following a Gaussian distribution:

$$p(x_{t+1}^{(i)}|\mathbf{x}_{<t}, z_t^{(ij)}) = \mathcal{N}(x_{t+1}^{(i)}|\mu_{t+1}^{(i)}, \sigma^2 I) \tag{3}$$

$$\mu_{t+1}^i = f_{\text{dec}}(\sum_{j \neq i} \sum_k z_{t,k}^{(ij)} u_k; \theta), \quad u_k = f_{\text{mlp}}^k(x_t^{(i)}, x_t^{(j)}) \tag{4}$$

Here the output $x_{t+1}^{(i)}$ is reparameterized by a Gaussian distribution with mean $\mu_{t+1}^{(i)}$ and a fixed standard deviation $\sigma^2$. The mean vector $\mu_{t+1}^{(i)}$ of agent $i$ is computed by aggregating the hidden states of all other agents. We use a separate MLP to encode the previous inputs into different type of edges in a k-dimensional one-hot vector $z_t^{(ij)}$. To generate long-term predictions using the model in Eqn. (4), we can also incorporate the predictions from the previous time step. The decoder architecture is the same as in NRI at a given time step, which consists of message passing GNN operations, followed by a GRU (Cho et al., 2014) decoder.

**Inference.** At every time step $t$, we learn a different distribution for the hidden relation $z_t^{(ij)}$. We assume a uniform prior for $p_\theta(\mathbf{z}_t)$ and use ELBO as the optimization objective:

$$\begin{aligned}
\mathcal{L}_{\text{ELBO}} &= \mathbb{E}[\log p_\theta(\mathbf{x}_{<T}|\mathbf{z}_{<T})] - \beta d_{\text{KL}}[q_\phi(\mathbf{z}_{<T}|\mathbf{x}_{<T})||p_\theta(\mathbf{z}_{<T})]] \tag{5} \\
&= -\sum_{i=1}^N \sum_{t=1}^T \frac{(\mu_t^{(i)} - x_t^{(i)})^2}{2\sigma^2} + \beta \sum_{i,j}^N \sum_{t=1}^T H(q_\phi(z_t^{(ij)}|\mathbf{x}_t))
\end{aligned}$$

where the mean vector $\mu_t^{(i)}$ is parameterized by the decoder. $H$ is the entropy function and $\beta$ balances the two terms in ELBO (Higgins et al., 2016).

## 4 EXPERIMENTS

We conduct extensive experiments on simulated physics dynamics and real-world basketball trajectories. The majority of our experiments are based on the physics simulation in the Spring environment. This is ideal for model verification and ablative study as we know the ground truth relations.

### 4.1 PHYSICS SIMULATIONS

**Data Generation**  The Spring environment (Kipf et al., 2018) simulates the movements of a group of particles connected by a spring in a box. The hidden relation is whether there is a spring connecting the two particles. To simulate dynamic relations, we generate the trajectories by removing and adding back the springs following certain patterns. Figure 4 visualizes the trajectories resulting from such dynamic relations. Starting from the bottom, the two-particle trajectories appear as straight lines and bend in the middle due to the spring force, and return to straight lines after the removal of the spring.

We define the number of time steps between the change of relations as the *dynamic period*. The primary challenges for dynamic relation inference arise along two dimensions:

1. The shorter the dynamic period, the more frequent the relation changes. Hence, it becomes more difficult to infer relations with shorter dynamic period.

2. If the dynamic period itself also changes, then the task becomes much harder because the model also needs to adapt to

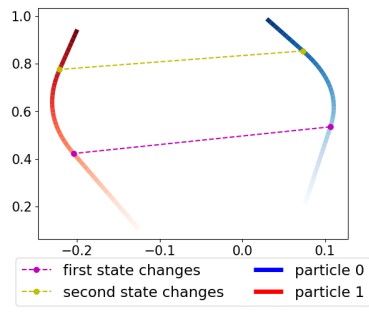

Figure 4: Example trajectories of two particles. with two relation changes. The trajectories start from the end with lighter color and gradually become darker.

the unknown period in the changing relations. Note that the way relations change in the trajectories must follow certain pattern and not be completely random. Otherwise it would be impossible to learn anything meaningful.

We experiment with two types of dynamic relations: periodic dynamics and additive dynamics. For periodic dynamics, we generate the trajectories by periodically removing and adding back the springs. We investigate the model performance by generating data with different frequencies of periodicity. For additive dynamics, we assume the dynamic period is increasing arithmetically. Each trajectory is of length 50 and the decoding length is 40, see details of the generated dataset in Appendix.

**Baselines and Setup**  We consider several baselines for comparison: (1) NRI (static): unsupervised NRI with an encoder trained using the entire trajectory and infer repeatedly over time. This corresponds to NRI (learned) in (Kipf et al., 2018). (2) NRI (adaptive): NRI (static) with an encoder trained over sub-trajectories. The encoding length corresponds exactly to the dynamic period of the dataset. We use the NRI (static) decoder to predict the entire trajectory in an auto-regressive fashion. (3) Interaction Networks (IN) (Battaglia et al., 2016): a supervised GNN model which uses the ground truth relations to predict future trajectories. We include this supervised learning model as the "gold standard" for our inference tasks. It is important to note here that (Graber & Schwing, 2020) also propose a model, dNRI, for this problem but the focus of their work is trajectory prediction whereas we focus on unsupervised relational inference. In our experiments with their model, we observed a relational inference accuracy of 0.505 on our periodic data with dynamic period 20. On the other hand, the same model gives an accuracy of 0.66 on the 3-particle synthetic data presented in their paper. Therefore, dNRI is unable to infer relations in an unsupervised setting.

In practice, we do not know the dynamic period beforehand. Therefore, how often we infer the relations is a difficult choice: rare inference would miss the time steps where relations change while predicting too frequently introduces more latent variables and complicates the inference. To investigate this trade-off, we define *inference period* as the number of time steps between two

predicted relations. Unless otherwise noted, the inference period in our experiments is the same as the dynamic period. All the models are trained to predict the sequence in an auto-regressive fashion: the prediction of the current time step is fed as the input to the next time step. We use Adam (Kingma & Ba, 2014) optimizer with learning rate $5e^{-4}$ and weight decay $1e^{-4}$ and train for 300 epochs.

### 4.1.1 PATHOLOGICAL CASES OF NEURAL RELATIONAL INFERENCE

It is known that latent variable models suffer from the problem of identifiability (Koopmans & Reiersol, 1950), which means certain parameters, in principle, cannot be estimated consistently. NRI infers correlation-like relations between trajectories which highly depend on the length of the time lag. To test this hypothesis, we follow the exact same setting as Kipf et al. (2018) to infer the interaction graph. Instead of decoding 50 time steps, we vary the length of input and output sequence.

Table 1 summarizes the inference accuracy with different sequence length in the encoder and decoder. We can see that the performance of NRI deteriorates drastically with shorter training sequences, simply increasing the capacity of the encoder (NRI++) does not help. One plausible explanation is that NRI is learning correlation-like interactions. Shorter decoding sequences carry less information about correlations, making it harder to learn. Meanwhile, we also observed that using auto-regressive can achieve better inference accuracy compared to teacher forcing. The pathological cases highlight the issue of NRI for dynamic relational inference. If the interactions change frequently every few time steps, repeatedly applying NRI to different time steps would suffer from short decoding sequences. Therefore, having a model that can jointly infer a sequence of relations is critical.

Table 1: Inference accuracy (%) of NRI trained with trajectory lengths. Note that the performance deteriorates significantly when the output length decreases. For NRI++, we added two more hidden layers to the MLP encoder of NRI.

| | Teacher Forcing | | | | Auto-regressive | | | |
|---|---|---|---|---|---|---|---|---|
| Output Length | 40 | 20 | 8 | 4 | 40 | 20 | 8 | 4 |
| NRI | 0.99 | 0.65 | 0.63 | 0.54 | 0.99 | 0.81 | 0.80 | 0.69 |
| NRI++ | 0.99 | 0.66 | 0.63 | 0.53 | 0.99 | 0.80 | 0.80 | 0.70 |

### 4.1.2 DYNAMIC RELATIONAL INFERENCE COMPARISON

We compare the performance of different models for dynamic relational inference tasks.

**Periodic dynamics** In the periodic scenario, the dynamic period is fixed. We generate four datasets with a dynamic period of $40, 20, 8, 4$ to simulate relational dynamics with increasing frequency. Table. 2 columns "40, 20, 8, 4" show the trajectory prediction mean square error (MSE) and interaction inference accuracy comparison of different methods.

We can see that all methods can achieve almost perfect predictions of the trajectories with very low MSE. However, there is a sharp difference in relational inference accuracy. NRI (static) is incapable of learning dynamic interactions. NRI(adaptive) can learn but has lower accuracy due to short decoding sequences. With a more expressive encoder and joint decoding, DYARI is able to reach higher accuracy. When the dynamic period is very small at 4, even DYARI struggles slightly, suggesting the fundamental difficulty with frequently changing dynamics.

Table 2: Performance comparison for ours and the baselines in both the periodic (40,20,8,4) and additive (Add) dynamic scenarios. MSE is for trajectory prediction and Accuracy quantifies the dynamic relational inference performance.

| Dynamic Period | MSE ↓ | | | | | Accuracy ↑ | | | | |
|---|---|---|---|---|---|---|---|---|---|---|
| | 40 | 20 | 8 | 4 | Add | 40 | 20 | 8 | 4 | Add |
| NRI (static) | 2.2e-4 | 5.2e-3 | 2.7e-3 | 2.4e-3 | 3.6e-3 | 0.99 | 0.52 | 0.51 | 0.50 | 0.53 |
| NRI (adaptive) | 2.2e-4 | 2.7e-3 | 1.3e-3 | 5.9e-4 | 3.1e-3 | 0.99 | 0.81 | 0.80 | 0.69 | 0.81 |
| DYARI | **2.6e-5** | **4.1e-5** | **4.6e-6** | **3.6e-6** | **7.6e-6** | **0.99** | **0.92** | **0.91** | **0.74** | **0.87** |
| IN | 2.9e-5 | 2.3e-5 | 4.3e-5 | 4.7e-5 | 3.9e-5 | 0.99 | 0.99 | 0.99 | 0.98 | 0.99 |

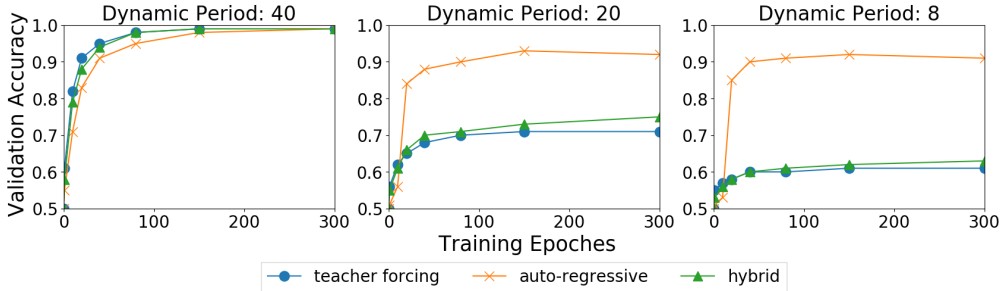

Figure 5: Learning curve of DYARI trained with teacher forcing (blue), auto-regressive (yellow), and hybrid (green) in decoder. Hybrid model is first trained with teacher forcing in the beginning 30 time steps and then auto-regressively in the later 10 time steps. We report the relational inference accuracy on the validation data for different dynamic periods.

**Additive Dynamics**    In the additive scenario, we allow the dynamic period itself to increase arithmetically. We increase the dynamic period in steps of 4 starting from a dynamic period of 4 timesteps. In a sequence of 40 timesteps, this implies that the relations (spring connection) get flipped at timesteps 4, 12 and 24. We use four NRI(static) models, each trained separately with 4, 8, 12 and 16 encoding timesteps. We combine the ensemble model predictions into NRI(adaptive). Table. 2 "Add" columns show the performance comparison. Similar to the periodic scenario, DYARI outperforms the baselines in this challenging task as well. Note that NRI(adaptive) is a close competitor w.r.t inference accuracy, but it is a four model ensemble and takes a long time to train.

### 4.1.3    ABLATIVE STUDY

We perform ablative study to further validate our experiment design and understand the behavior of DYARI. In particular, we study the trade-off between dynamic and inference period, the effect of training scheme, as well as the ablative study of model architecture design.

**Dynamic vs. Inference Period.**    To understand the relations between dynamic and inference period, we repeat the periodic scenario experiments by varying both dynamic and inference period in $40, 8, 4$ time steps.

In Table 3, we observe that dynamic relational inference reaches the highest accuracy when the inference period matches the dynamic period. If the inference period is longer than the dynamic period, the model can miss the changes in the relations and completely fails to perform inference. Meanwhile, if the inference period is shorter than the dynamic period, the model still can learn but suffers from low accuracy. This is potentially due to the extra uncertainty introduced by estimating more latent variables.

Table 3: Inference accuracy for different combinations of dynamic and inference periods with DYARI.

| **Dynamic Period** | 40 | 8 | 4 |
|---|---|---|---|
| **Inference Period** 40 | **0.99** | 0.50 | 0.50 |
| 8 | 0.88 | **0.80** | 0.50 |
| 4 | 0.80 | 0.76 | **0.74** |

**Decoder Training Scheme**    Another fundamental challenge in sequence prediction is covariate shift (Bickel et al., 2009) – a mismatch between distribution in training and testing – due to sequential dependency. Common solutions to mitigate covariate shift include teacher forcing (Williams & Zipser, 1989) and scheduled sampling (Bengio et al., 2015). However, all these work are focused the prediction of *observed* sequence while our sequence predictions are on the *latent* variables. It is not evident that covariate shift exists in this setting. We demonstrate the empirical evidence for the effect of different training schemes on the accuracy of relational inference.

Quite surprisingly, we found that auto-regressive training is most effective for dynamic relations inference. Figure. 5 summarizes the difference in learning curve between using teacher forcing and auto-regressive for different dynamic periods. We also include a version of scheduled sampling (hybrid): in the first 30 time-steps, we train the model with teacher forcing and then switch to auto-regressive in the last 10 time-steps. We observe that while teacher forcing converges faster, it

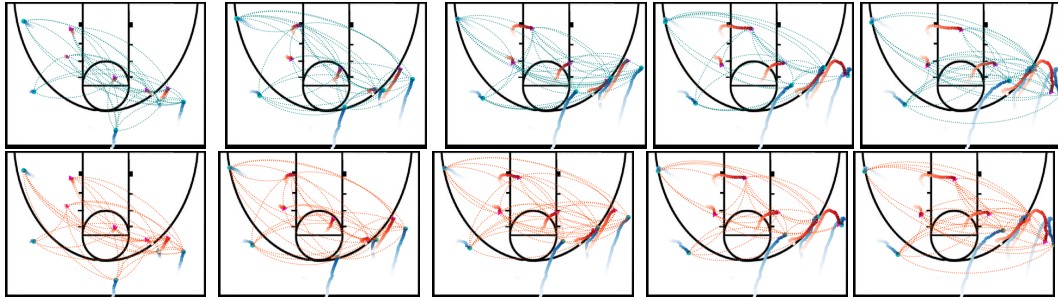

Figure 6: Visualization of the inferred relations (dashed links) in the basketball players trajectories over time by DYARI with an inference period of 8. The blue dashed links in the top are the inferred interactions from the same team (coordination) and red dashed links in the bottom are from different teams (competition). Different columns represent different time steps.

| | MSE ↓ | | | | NELBO ↓ | | | |
|---|---|---|---|---|---|---|---|---|
| **Inference Period** | 40 | 20 | 8 | 4 | 40 | 20 | 8 | 4 |
| NRI(static) | 2.3e-3 | - | - | - | 13.71 | - | - | - |
| NRI(adaptive) | 2.3e-3 | 3.0e-2 | 3.3e-2 | 9.7e-3 | 13.71 | 303.10 | 337.54 | 96.76 |
| DYARI | 2.2e-3 | 8.4e-4 | 4.6e-4 | 1.8e-4 | 12.65 | 6.16 | 4.38 | 3.67 |

Table 4: Performance comparison for DYARI and baselines on the real-world basketball trajectory dataset with different inference periods 40, 20, 8 and 4.

leads to lower accuracy. This observation is consistent across different dynamic periods. Therefore, auto-regressive training is preferred for dynamic relation inference.

## 4.2 REAL-WORLD BASKETBALL DATA EXPERIMENTS

To showcase the practical value of dynamic relational inference, we apply DYARI to a real-world basketball trajectory dataset. The goal of the experiment is to extract meaningful "hidden" relations in competitive basketball plays. The basketball dataset contains trajectories for 10 players in a game. As the ground-truth relations are unknown, we use the trajectory prediction MSE and negative ELBO as in-direct measures for the dynamic relational inference performance. We assume there are two types of hidden relations: coordination and competition. We defer the details of the dataset and training setup to the Appendix.

We report performance comparisons for different inference periods. As shown in Table 4, we observe lower MSE loss and negative ELBO with shorter inference period. Intuitively, the interactions in the real world may change constantly, thus shorter inference period can capture the dynamics better. DYARI outperforms the baselines in trajectory prediction MSE and negative ELBO loss. Notice that NRI(adaptive) is using encoder and decoder that are trained separately and this results in a high negative ELBO loss on the test set. Fig. 6 visualizes a sample trajectory of 10 basketball players with inferred relations from DYARI over different time steps. We separate coordination and competition interactions in different rows. In Fig. 6, Kobe Bryant is moving along with three-point line and guarded by a defender. We can see clear attention drawn to the specific players throughout the play. See Appendix for other inferred relations.

## 5 CONCLUSION

We investigate unsupervised learning of dynamic relations in multi-agent trajectories. We propose a novel deep generative model: Dynamic multi-Agent Relational Inference (DYARI) to infer changing relations over time. We conduct extensive experiments using a simulated physics system to study the performance of DYARI in handling various dynamic relations. We perform ablative study to understand the effect of dynamic and inference period, training scheme and model design choice. Compared with static NRI and its variant, our DYARI model demonstrates significant improvement in simulated physics systems as well as in a real-world basketball trajectory dataset.

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

## A    MODEL IMPLEMENTATION DETAILS

In this section, we include some details about the model implementation, especially the encoder part. Our encoder is analogical to ResNet CNNs (He et al., 2016) used in the field of image recognition, where the task can be abstracted to be a classification problem on 1D dimension. Meanwhile, inspired by PSPNet used in visual scene semantic parsing (Zhao et al., 2017), we add additional 2 global feature extractors to combine the whole-sequence (global) features and the sub-sequence (local) features.

for DYARI, each residual block shown in Fig. 3 consists of 4 skip connections structure.

```python
def conv_1d(in_planes, out_planes, kernel_size=3, stride=1):
    """1 dimensional convolution with padding"""
    return nn.Conv1d(in_planes, out_planes, kernel_size=kernel_size, stride=1,
                     padding=kernel_size//2, bias=True)

class BasicBlock(nn.Module):
    expansion = 1

    def __init__(self, inplanes, planes, kernel_size=3, stride=1, downsample=None):
        super(BasicBlock, self).__init__()
        self.conv1 = conv_1d(
            inplanes, planes, kernel_size=kernel_size, stride=stride)
        self.bn1 = nn.BatchNorm1d(planes)
        self.relu = nn.LeakyReLU(inplace=True)
        self.conv2 = conv_1d(
            planes, planes, kernel_size=kernel_size, stride=stride)
        self.bn2 = nn.BatchNorm1d(planes)
        self.downsample = downsample
        self.stride = stride

    def forward(self, x):
        residual = x

        out = self.conv1(x)
        out = self.bn1(out)
        out = self.relu(out)

        out = self.conv2(out)
        out = self.bn2(out)

        if self.downsample is not None:
            residual = self.downsample(x)

        out += residual
        out = self.relu(out)

        return out
```

Figure 7: Pytorch code snippet of the Residual Block used in DYARI encoder.

## B    EXPERIMENTAL DETAILS

**Particle dataset**    In general, we use the same pre-processing in NRI. Each raw simulated trajectory has length of 5000 and we sample with frequency of 100 so that each sample has length of 50 in our dataset. Correspondingly, the value of dynamic period/inference period matches the length of sample in our dataset. For instance, dynamic period = 10 means that the in the raw trajectory, the state of a node changes every 1000 time steps. In addition, The value of trajectories are all normalized to range of $[0, 1]$ and the evaluation is done on the same range as well.

**Basketball dataset details**    The basketball dataset consists of trajectory from 30 teams. The raw trajectory is captured with frequency of 25 ms. For our experiment, we sample the trajectory with frequency of 50 ms for more evident player movements. We use a inference period that matches the length of sample. For instance, inference period = 10 means that our model produce prediction every 500 ms. The resulting dataset include 50,000 training samples, 10,000 validations samples and 10,000 test samples.

We normalize the values of the trajectories to range [0,1] and train all the models in an auto-regressive fashion. We use the same training set up as in physics simulation experiments with a batch size of 64.

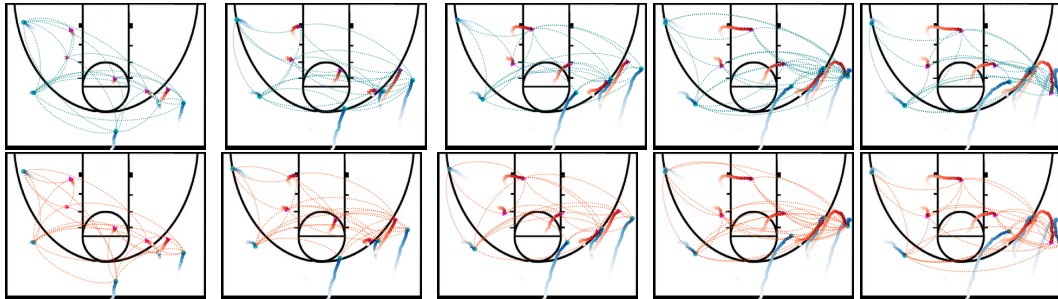

Figure 8: Visualization of the basketball players trajectories with inference period = 8. The top row visualizes the inferred interactions from the same team (coordination) and the bottom row visualizes the inferred interactions from different teams (competition). Different columns represent different time steps.

## C  ADDITIONAL EXPERIMENTS

**Stochastic dynamics**  In order to make the problem even harder and to unify all the previous settings, we generate a dataset where the edge types are flipped randomly with a probability $p$ after each dynamic period of 4 timesteps. The static data generation corresponds to $p = 0$ and the periodic dynamics corresponds to $p = 1$. Table 5 shows the MSE and inference accuracy of NRI, DYARI and Interaction Networks on the stochastic dataset for flipping probabilities $p = 0.8$ and $p = 0.9$.

Table 5: Qualitative results for stochastic dynamics. Accuracy improves by increasing the model capacity. In the training, The inference period of the two DYARI match with the dynamic period.

|  | MSE | | Accuracy | |
| --- | --- | --- | --- | --- |
| **Flipping Probability** | 0.8 | 0.9 | 0.8 | 0.9 |
| NRI | 1.4e-3 | 2.3e-3 | 0.59 | 0.60 |
| DYARI | 8.3e-4 | 1.8e-3 | 0.57 | 0.63 |
| IN (Supervised) | 4.5e-5 | 4.2e-5 | 0.99 | 0.99 |

Table 6: Results with and without average pooling in the interpolation module of DYARI.

|  | MSE | Accuracy |
| --- | --- | --- |
| DYARI without average pooling | 1.8e-5 | 0.59 |
| DYARI with average pooling | 4.1e-5 | 0.92 |

**The Effect of Average Pooling**  We perform an ablation study where we remove the average pooling corresponding to the inference period in the interpolation module to study the effect of this average pooling on the results. We find that without average pooling, inference accuracy decreases from 0.92 to 0.59 (as shown in Table 6) for inference period at 20. Here we drop the average pooling corresponding to the inference periods and directly interpolate to the sequence length. This shows that intermediate average pooling is critical for the relational inference performance.

**Additional Inferred Relations in Basketball Trajectories**  We set the number of relations in Basketball dataset as two. In Sec 4.2, we visualized one of the inferred relations. Table 8 visualizes the second relations. Notice that the first relation captures focus on the rightmost red player while here the relation captures focus on the leftmost red player.

