# OpenReview forum: "Dynamic Relational Inference in Multi-Agent Trajectories"
_ICLR.cc/2021/Conference — Reject_

### Official Review · AnonReviewer1 · 2020-10-25
**Extension of Neural Relational Inference for dynamic relations**

**Rating:** 2
**Confidence:** 5

**Review:**

This paper introduces DYnamic multi-Agent Relational Inference (DYRI), which is an extension of Neural Relational Inference (NRI) for dynamic relations.  This extension shows improved performance on the real-world basketball trajectory dataset.

The article is fairly well structured, apart from the literature review, which is somehow missed very important and very related works. The paper is clearly motivated and seems to be technically correct.

My main concern with this paper is a novelty. The main argument is to increase the expressive power of static NRI for dynamic relations, which I think is a very valid motivation for this paper; however, this idea has been already published as Dynamic Neural Relational Inference (dNRI; Graber & Schwing, CVPR 2020). Based on my understanding, the model and inference of DYRI are the same as dNRI.  The authors have to clearly discuss their difference with dNRI if they still think those are different. Also, I would have expected to see it as a baseline.

As I mentioned above, one of the main drawbacks of the paper is its literature review and baseline methods. Besides dNRI, this problem has been tackled in other papers with different structures. For example, Graph-VRNN (Chen Sun, ICLR 2019) combined variational RNNs and NRI for multi-agent interactions and applied it in the same basketball dataset. Also, Dynamic Neural Relational Inference for Forecasting Trajectories has been proposed in CVPR 2020.

---

### Official Review · AnonReviewer2 · 2020-10-26

**Rating:** 4
**Confidence:** 4

**Review:**

This paper presents a method for dynamic relational inference for multi-agent trajectory prediction. The method extends the neural relational inference (NRI) (Kipf et al., 2018) by changing the static relations between agents to dynamic relations. This equates to inferring time-varying latent variables $z_t^{ij}$ as opposed to learning time-independent latent variables $z^{ij}$. The paper conducts experiments on physics simulations and basketball trajectories to show the superiority of the proposed method against different variants of NRI.


**Pros:**
++ The paper is well motivated and clearly written.
++ The paper performs detailed experiments and ablation studies to analyze the behavior of the proposed model and the pitfalls of NRI.


**Concerns:**
-- My major concern of this paper is that the almost same idea has been proposed in prior work [1]. The probabilistic formulation in that paper is essentially the same as this paper, which also introduces time-varying latent variables $z_t^{ij}$ and infers them within a VAE model. This drastically reduces the novelty of this paper. [1] also performs more extensive experiments on simulation, human motion, basketball, and traffic trajectory.

-- My second concern is about the design “the inference period in our experiments is the same as the dynamic period”. This is a rather strong assumption, which cannot be realized in real applications since the dynamics period is not known. This assumption also seems to defeat the purpose of dynamic relational inference, which is to infer the change in dynamics so the model should be able to have a much longer inferior period than the dynamics change period. From table 3, it becomes evident that the model only performs well when the inference period is shorter than the dynamics period, which means when the relationship between agents stays unchanged during a single inference period.

-- The decoder distribution in Eq. 2 should have multiple $p(x_{t+1}^{(i)}|x_{<t}^{(i)}, z_{t}^{(ij)})$ for different $j$. In the current form, it seems only one $j$ is used and not clear which one.

[1] Dynamic Neural Relational Inference. Graber et al. CVPR 2020.

**Rating Justification:**
A very similar idea to this paper has been proposed in prior work [1], which significantly reduces the novelty of this paper. There are also concerns about the model’s design (e.g., inference period). Thus, I vote for a reject.

**Additional Comments:**
-- the notation $<t$ is used to denote observations that also include t, which is quite confusing. $\leq t$ is probably better.
-- Fig 2 does not match Eq 2. Since the $x_2^{(1)}$ is generated by  $z_2^{(12)}$ and $z_2^{(13)}$, so $z_t^{ij}$ in Eq 2 should be $z_{t+1}^{ij}$ to match the time index of $x_{t+1}^{(i)}$

---

### Official Review · AnonReviewer4 · 2020-10-27
**Two contributions that should be highlighted and placed in the context of existing methods separately.**

**Rating:** 5
**Confidence:** 5

**Review:**

The authors propose a novel Relational Inference system that learns to predict the graph structure underlying the data as well as the updated state of the system. Relational reasoning has received considerable attention in recent year. Predicting the graph structure underlying a system from data in a dynamic way is an great next step, which could help alleviate some of the scalability issue currently afflicting these methods.

The way I read this paper, there are two distinct high level contributions:
1. Dynamically infer the graph structure underlying a trajectory, from data and use it to predict the next step
2. A specific method to accomplish the first contribution.

It is encouraging to see that DYARI outperforms NRI in the cases presented. However, it is possible to imagine a dynamic version of NRI (where the inferred graph structure is updated every time-step after a burn-in period).
Moreover, while NRI is a strong baseline, other similar methods have been proposed to tackle relational inference e.g. Graph Attention Networks, Transformers and even straight Graph Networks. Similarly to NRI, with a simple thresholding mechanism these could be used to predict the graph structure as well, both statically and dynamically.

It would be really helpful to better place this work in the context of existing methods and to try and tease out the two contributions. Is it a good idea to dynamically infer the graph structure, and if so should we use this specific method?

Finally, predicting the graph structure lends itself to really interesting analysis, and the dynamic aspects of this are completely unexplored. Imagine finding that your method infers that particle clouds form disjoint objects that become connected when they collide, wouldn't that be something?

With a broader outlook, we should note that Relational Inference and Graph-based models have received a lot of attention in recent years. Methods have scaled to thousands of nodes and very complex physics (see e.g. Learning to simulate complex physics with graph networks). The datasets reported in this work include up to 10 interacting objects. Since the dynamic inference of the connectivity structure is especially apt to larger scale experiments, it would be really great to see results on more complicated datasets that include computational cost considerations when compared to static graph baselines, although this might be outside the scope of this initial paper.

Thank you very much for sharing these cool ideas. I genuinely believe this is a research direction worth investigating. I look forward to our discussion.

All the best!

---

### Official Review · AnonReviewer3 · 2020-10-29
**Improvement on NRI but confusing and limiting design choices**

**Rating:** 4
**Confidence:** 3

**Review:**

This paper builds on Kipf et al. (2018)’s Neural Relational Inference. In particular, this work introduces a latent variable model which treats the interactions (i.e. relations) between different agents as dynamic and time-varying. As in NRI, the interaction variable between any two agents is conditioned on the history of those agents’ states. An agent’s future state is conditioned on its history of states as well as its interaction variables with other agents.

The results from DYARI are interesting but I am concerned about the intricacies of setting the “inference period” to be aligned with the “dynamic period.” I would have expected that choosing a smaller inference period (e.g. half the dynamic period) should lead to little to no loss in performance. The authors ascribe the observed loss in performance to “the extra uncertainty introduced by estimating more latent variables.” I’m not totally convinced.

Physical systems like springs have an inherent temporal invariance which you don’t seem to be exploiting. Did you consider (a) processing timesteps in a moving window of size T (i.e. z^{ij}_t is conditioned only on the last T observations) or (b) using a recurrent encoder to process observations sequentially, rather than process all timesteps at once with the PSPNet encoder? That might help with the inference challenge you're facing.

More questions and requests for clarification:
- What if the inference period is out of sync with the dynamic period (e.g. 3 steps versus 5 steps)? Does the model still perform reasonably?
- Why not model relations as continuous latents (setting aside the fact that NRI used discrete variables)? That way you can model negative interaction (e.g. competition), zero interaction, as well as the magnitude of interaction.
- I don’t understand Figure 8 at all. Consider the two blue players who stay on the leftmost side of the basketball court. They have a dotted blue line between them in all the “coordination” plots and a dotted red line between them in all the “competition” plots. Could you please explain what's going on? I would expect only one relation to be on at any time step.
- In Table 6, you show average pooling leads to higher accuracy but also a higher MSE. Shouldn't you have a lower MSE with a higher accuracy?
- Could you please add error bars for your reported scores over independent runs?
- What value of Beta did you use for your loss? What was the effect of varying this hyperparameter?

Minor:
- In equation 2, I assume the density p(x^(i)_{t+1} | x^(i)_{< t}, z^{ij}_t) was meant to be conditioned on N-1 latents not just a single j? In fact, j is undefined in the equation at that point.
- Figure 2 shows the hidden interaction nodes z^{ij}_t are not conditioned on any other nodes. That is not true from equation 2: each z^{ij}_t seems conditioned on agent i and j’s past trajectories.
- Could you please use x_{\leq t} instead of x_{< t} to denote states up to and including time t?

---

### Decision · Program_Chairs · 2021-01-07
**Final Decision**

**Decision:**

Reject

**Comment:**

This paper presents a method for relational inference in multi-agent/multi-object trajectory prediction tasks. Different from the neural relational inference (NRI) model [1], the presented method is able to model time-varying relations. Experimental results on physics simulations and sports games (basketball) show benefits over variants of the NRI model.

The reviewers agree that the presented method is mostly solid, that the experiments are insightful, and that this is generally a well-written paper. The authors, however, have apparently overlooked recent related work [2] (dNRI) that proposes a very similar model. In the light of dNRI, it is difficult to argue for the novelty of the presented approach, and the paper needs to undergo a revision in order to more clearly differentiate it from the dNRI model, and to resolve the other concerns raised by the reviewers.

[1] Kipf et al., Neural Relational Inference for Interacting Systems (ICML 2018)
[2] Graber et al., Dynamic Neural Relational Inference (CVPR 2020)